# IL-2 and IL-1β Patient Immune Responses Are Critical Factors in SARS-CoV-2 Infection Outcomes

**DOI:** 10.3390/jpm12101729

**Published:** 2022-10-17

**Authors:** Shereen Fawzy, Mesaik M. Ahmed, Badr A. Alsayed, Rashid Mir, Dnyanesh Amle

**Affiliations:** 1Department of Medical Microbiology, Faculty of Medicine, University of Tabuk, Tabuk 71491, Saudi Arabia; 2Department of Internal Medicine, Faculty of Medicine, University of Tabuk, Tabuk 71491, Saudi Arabia; 3Department of Medical Laboratory Technology, Faculty of Applied Medical Sciences, University of Tabuk, Tabuk 71491, Saudi Arabia; 4Department of Biochemistry, All India Institute of Medical Sciences, Nagpur 441108, India

**Keywords:** IL-2, IL-1β, cytokine storm, COVID-19, immune response, morbidity, mortality

## Abstract

Background: Immune dysregulation has been linked to morbidity and mortality in COVID-19 patients. Understanding the immunology of COVID-19 is critical for developing effective therapies, diagnostics, and prophylactic strategies to control the disease. Aim: The aim of this study was to correlate cytokine and chemokine serum levels with COVID-19 disease severity and mortality. Subjects and Methods: A total of 60 hospitalized patients from the Tabuk region of Saudi Arabia with confirmed COVID-19 were included in the study. At hospital admission, the IL-1 β, IL-2, IL-8, IL-10, LT-B4, and CCL-2 serum levels were measured. The cytokine levels in COVID-19 patients were compared to the levels in 30 healthy matched control subjects. Results: The IL-1 β, IL-2, LTB-4, CCL-2, and IL-8 levels (but not IL-10) were significantly higher in all COVID-19 patients (47 survivors and 13 non-survivors) compared with the levels in the healthy control group. In the non-survivor COVID-19 patients, patients’ age, D-dimer, and creatinine kinase were significantly higher, and IL-1 β, IL-2, and IL-8 were significantly lower compared with the levels in the survivors. Conclusion: Mortality rates in COVID-19 patients are associated with increased age and a failure to mount an effective immune response rather than developing a cytokine storm. These results warrant the personalized treatment of COVID-19 patients based on cytokine profiling.

## 1. Introduction

The emergence of severe acute respiratory syndrome coronavirus-2 (SARS-CoV-2) in December 2019 resulted in a coronavirus pandemic. By the end of March 2022, 750,910 confirmed cases of COVID-19 and 9047 deaths had been reported in Saudi Arabia [1]. The major components of COVID-19 pathogenesis are pulmonary inflammation with significant lung damage and generalized immune dysregulation, and COVID-19-associated morbidity and mortality have been linked to an altered host immune response [2]. Two competing hypotheses explain the severity and mortality of COVID-19: a cytokine storm and the failure of the host to mount a protective immune response, resulting in uncontrolled viral dissemination and organ injury.

The cytokine storm was considered the most likely explanation for the deterioration and progression of severe COVID-19 cases for respiratory failure and death. The cytokine storm can cause viral sepsis and inflammatory-induced lung injury, where pneumonitis, acute respiratory distress syndrome, respiratory failure, shock, multiorgan failure, and death are possible complications [3]. COVID-19 patients have high levels of proinflammatory cytokines, including interleukin (IL)-1β, IL-6, IL-8, interferon-gamma (IFNγ), 10 kD interferon-gamma-inducible protein (IP-10), and monocyte chemoattractant protein-1 (MCP-1, CCL-2), which probably activate T-helper-1 (Th1) cell responses [4]. Furthermore, patients who required intensive care unit (ICU) admission had higher levels of granulocyte colony-stimulating factor (G-CSF), IP-10, MCP-1, macrophage inflammatory protein 1 (MIP-1), IL-6, and tumor necrosis factor (TNF) than patients who were not admitted to the ICU [5]. The concentrations of granulocyte colony-stimulating factor (G-CSF), IL-2, IL-7, IL-10, TNF, and the chemokines, CCL2, CCL3, and CXCL10 were extremely high in the plasma of patients with severe COVID-19 [6].

Paradoxically, the second theory for the morbidity and mortality caused by COVID-19 is an “immunologic collapse” of the host’s immune system, manifested as unrepressed viral replication and spreading with direct host cytotoxicity. This opposing theory is supported by progressive and profound lymphopenia, which is comparable to the levels in AIDS patients. Several lymphocyte subsets, including CD4⁺ T, CD8⁺ T, and natural killer cells, which play important antiviral roles, and B cells, which are required for viral-neutralizing antibody production, are lost [7]. IL-2 deficiency appears to be a critical factor contributing to severe manifestations of COVID-19 infection, such as acute respiratory distress syndrome, multiorgan failure, and death [8].

Divergent immunotherapy avenues have emerged based on these two divergent hypotheses. Systemic corticosteroids are typically the first line of treatment to control the severity of systemic inflammation [9]. However, while initially beneficial, corticosteroid therapy has been linked to increased viral loads [10]. Widespread inflammation develops in patients with COVID-19. Therefore, inhibitors of proinflammatory cytokine signaling may have therapeutic value [11]. On the other hand, recombinant interleukin-2 stimulates lymphocyte recovery in patients with severe COVID-19 [12]. Age, obesity, pregnancy, and multiple comorbidities are significantly associated with worse outcomes in patients with COVID-19. In addition to the immunological response, other parameters were linked to the cytokine storm and mortality risk, including creatine kinase (CK), lactose dehydrogenase (LDH), C-reactive protein (CRP), and D-dimer.

The effectiveness of antiviral therapies is unclear, and national COVID-19 management protocols are updated based on the available literature [13]. Thus, understanding the immunology of COVID-19 is critical for developing effective therapies and diagnostic and prophylactic strategies for this disease. Our study aimed to develop a predictive model of COVID-19 mortality based on patient clinical features and immunological responses to guide diagnostic and therapeutic intervention and efficiently utilize the available resources during the pandemic.

## 2. Subjects and Methods

### 2.1. Patients and Controls

A total of 60 patients who were diagnosed with COVID-19 and 30 healthy age- and gender-matched control subjects were included in the study. The COVID-19 diagnosis was confirmed by RT-PCR using nasopharyngeal and oropharyngeal samples from all study participants. All eligible participants were enrolled consecutively during the study. Patients under the age of 18 or taking immunosuppressive or immunomodulatory drugs (corticosteroids and cytokine antagonists) were excluded from the study.

### 2.2. Study Design and Sample Collection

The following data were collected at admission: demographic, clinical characteristics, radiological findings, laboratory values for CRP, ferritin, D-dimer, and CK, and routine laboratory tests, including hemogram and liver and renal function tests. Patients were followed up prospectively with a standardized clinical approach until hospital discharge or demise. In addition to the routine tests, 3–5 mL of peripheral blood was collected from patients on admission and from the healthy control patients. Serum was separated from whole blood by centrifugation at 2500 rpm for 15 min and stored at −80 °C. Patients were clinically followed, and outcomes (improvement or death) were determined for the hospital stay period.

### 2.3. ELISA and Cytokine Measurements

Serum cytokine and chemokine levels were measured using commercial ELISA assay kits (IL-1, IL-2, LT-B4, and CCL-2, Abbkine Scientific Co. Ltd., Wuhan, China; IL-8 and IL-10, KRISHGEN Biosystems, Mumbai, India), following manufacturers’ instructions. Serial two-fold dilutions of standards were prepared from initial concentrations of 250, 125, 1000, 500 pg/mL, and 60 ng/L and 2000 pg/mL for IL-1, IL-2, IL-8, IL-10, LT-B4, and CCL-2, respectively. Five dilutions were used to generate standard curves.

### 2.4. Statistical Analyses

Data are presented as frequencies (percentages), means ± standard deviations, or medians (interquartile ranges), as appropriate. Linearity of all quantitative data was assessed using Kolmogorov Smirnov analyses and groups were compared using Student’s unpaired *t*-tests, Mann–Whitney-U tests, or Kruskal–Wallis tests followed by post hoc Dunnet’s tests as appropriate for the type of data. ROC curve analyses were used to assess the ability of various parameters at admission to predict mortality. Odd’s ratios were calculated considering 95% confidence intervals. SPSS software, version 19.0 (Statistical Package for the Social Sciences Inc., Chicago, IL, USA) was used for all statistical analyses. A significance level of <0.05 was considered statistically significant.

## 3. Results

The 60 patients were grouped based on their outcomes. A total of 47 (78.33%) subjects survived and improved, and 13 (21.67%) succumbed. The comparisons of clinical characteristics are shown in Table 1. Age was significantly higher in the non-survivor group compared to age in the survivor group (*p* = 0.002). No significant differences in the frequency of symptoms were noted between the two study groups. A history of diabetes mellitus (Odd’s ratio: 6.25 95% C.I.:1.24–31.32, *p* = 0.015) and ischemic heart disease (Odd’s ratio: 3.625, C.I. 0.972–13.52, *p* = 0.047) were significantly more common in the non-survivors compared with the frequency in the survivors. Histories of smoking, hypertension, or chronic kidney disease were not significantly different between the two groups. The D-dimer (*p* < 0.0001) and CK (*p* = 0.015) levels were significantly higher in the non-survivors compared with the levels in the survivors. The treatments, immunomodulators, antiviral medications, and steroids, were similar between the two groups. However, significantly more non-survivors required oxygen therapy (*p* = 0.003), mechanical ventilation (*p* = 0.007), and ICU admission (*p* < 0.0001). The duration of hospital stay (*p* = 0.03) and O_2_ therapy (*p* = 0.019) were significantly longer in the non-survivors compared with the duration for the survivors.

The levels of IL-1 β and IL-2 were significantly lower in the non-survivors compared to the levels in the survivors (*p* = 0.001 and 0.011, respectively) (Table 2). LTB-4 also tended to be lower in the non-survivors compared to the survivors (*p* = 0.056). No significant differences were detected in the other measured cytokines and chemokines.

All cytokine and chemokine levels except for IL-10 were higher in both of the groups of COVID-19-positive subjects compared with the levels in the healthy controls. Post hoc analyses revealed that IL-1 β, IL-2, LTB-4, CCL-2, and IL-8 were significantly higher in COVID +ve subjects from both the survivor and non-survivor groups compared to the control group. IL-1 β, IL-2, and IL-8 levels were significantly lower in the non-survivors compared with the levels in the COVID +ve survivors (Table 3).

The predictive potentials of various parameters were assessed using ROC curve analysis. Age (sensitivity = 84.6% and specificity = 66.0% above 60 years), D-dimer (sensitivity = 92.3% and specificity = 66.0% above 151 ng/mL), and creatinine kinase (sensitivity = 84.6% and specificity = 61.7% above 142.5 U/L) predicted mortality. In addition, IL-1 β had the best area under the curve (80.2%) with a sensitivity of 84.6%. The IL-1 β sensitivity was 68.1% below the cut-off value of 109.46 (pg/mL). IL-2 (sensitivity = 84.6% and specificity = 70.2% below 85.54 pg/mL) also predicted poor prognosis (Table 4, Figure 1).

ROC curve analysis for the prognostic accuracy of the various parameters revealed that higher mortality risk was associated with increasing age and higher D-dimer and CK levels (Figure 1) and decreased levels of IL-1 β, IL-2, and LTB-4 (Figure 2).

## 4. Discussion

COVID-19 affected millions of people globally since it was declared a pandemic by the World Health Organization on March 11, 2020 [14]. Cytokine levels are important indicators of clinical disorders, and accurate quantification of cytokines provides valuable information for monitoring patient immune status in various diseases [15]. In this study, distinct cytokine and chemokine profiles and their correlation with disease outcome shed light on those factors linked with COVID-19 severity and mortality. In this study, IL-1 β, IL-2, LTB-4, CCL-2, and IL-8 levels were higher in all COVID-19 patients (survivors and non-survivors) compared with the levels in the healthy control group. However, succumbed patients generally showed lower levels of IL-1 β, IL-2, and IL-8 than the levels in patients who survived COVID-19. Furthermore, significant differences were detected between the non-survivors and survivors regarding age and the levels of D-dimer and CK; the values for the non-survivors were significantly higher than the values for the survivors. Lower levels of IL-1 β, IL-2, and LTB-4 and higher levels of D-dimer and CK and advanced age predicted a higher risk of death.

The results of this study confirm that immune system changes due to aging generally result in a decline in immunity to pathogens. The senescent immune system’s diminished capacity is clinically evident; aging is associated with high morbidity and mortality rates for various infections and lower vaccine efficacy in older patients [16]. T cells play an important role in the body’s response to COVID-19. Our results show that a multi-layered immune response is important for controlling the virus during the acute phase of infection and reducing disease severity. Evidence suggests that T cells play a more significant role than antibodies [17]. Niessl et al., reported that T cells are important for the viral clearance of SARS-CoV-2 and limit the severe cytokine storm associated with COVID-19 [18]. With aging, the senescent immune system’s supply of different T cell subsets decreases, and fewer cells are available to respond to a new virus. The Jolla Institute explained why older COVID-19 patients are much more vulnerable to the disease. With increasing age, the reservoir of T cells that can be activated against a specific virus decline, and the body’s immune response becomes less coordinated, contributing to an increased susceptibility to severe or fatal COVID-19 [17]. IL-2 deficiency appears to be a critical factor in SARS-CoV-2 infection, leading to more severe manifestations, such as a weak immune response to the virus, acute respiratory distress syndrome, multiorgan failure, and death [8]. This explanation is supported by our results showing that non-survivors express lower levels of IL-2, IL-1β, and IL-8 but higher levels of D-dimer and CK, denoting excessive tissue damage. The results from the study done by Remy et al., (2020) strongly suggest that the primary endotype of COVID-19 is one of immunosuppression rather than hyperinflammation. They concluded that COVID-19 causes a profound defect in host immunity rather than hypercytokinemia-induced organ injury, and the defect in host immunity includes the profound depletion of effector immune cells and severe functional defects in the T cells and monocytes [7]. Our results agree with these conclusions; COVID-19 patients with severe illness and poor prognosis had lower levels of IL-1B, IL-2, and IL-8 than COVID-19 patients with good prognosis. Thus, broadly inhibiting the host inflammatory response with steroids or biological inhibitors of the cytokines or cytokine receptors may worsen clinical outcomes in some COVID-19 patients by further impairing the compromised host’s protective immune response.

## 5. Conclusions

Understanding the immunological response to COVID-19 is critical for developing effective therapies and diagnostic and prophylactic strategies to control the disease. According to our results, different cytokine profiles are associated with COVID-19 severity and can differentially predict mortality. These results warrant the personalized treatment of COVID-19 patients based on cytokine profiling. Our study also confirms the effects of age on immunity to pathogens; aging is linked to high morbidity and mortality rates for various infections, as well as vaccine efficiency.

## Figures and Tables

**Figure 1 jpm-12-01729-f001:**
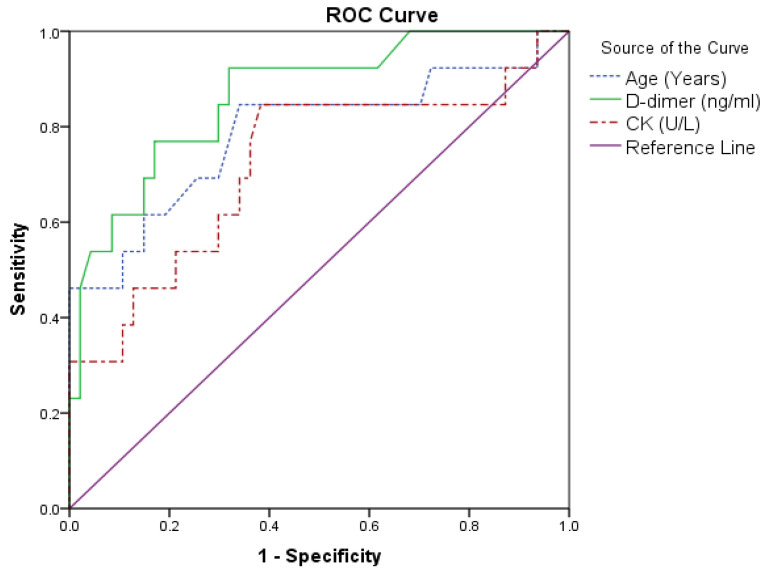
Parameters of the high-value predictors of mortality.

**Figure 2 jpm-12-01729-f002:**
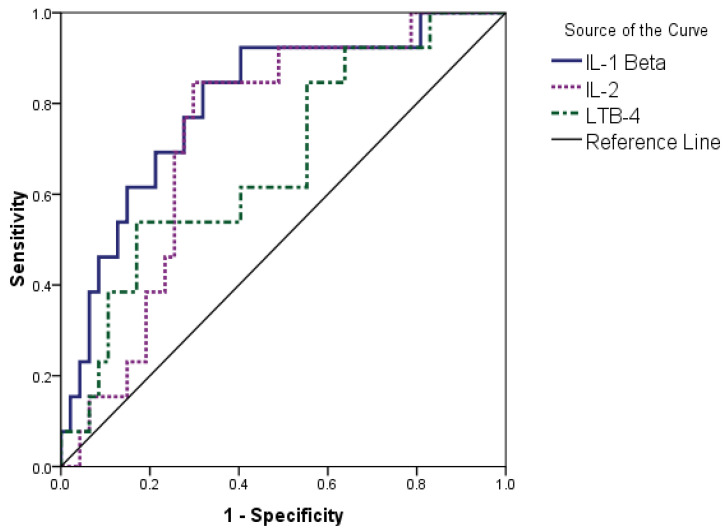
Parameters of the low-value predictors of mortality.

**Table 1 jpm-12-01729-t001:** Characteristics of study subjects.

Characteristics		Survivors (*n* = 47)	Non-Survivors (*n* = 13)	t/Mann–Whitney U Test Whitney U/*X*^2^	*p* Values
Age (years)		54.17 ± 13.77	69.15 ± 16.55	−3.321 ^#^	0.002
Gender	Male	22 (46.8)	09 (69.2)	2.56 *	0.27
Symptoms	Fever	28 (60.9)	08 (61.5)	0.002 *	0.614
	Cough	38 (82.6)	09 (69.2)	1.12 *	0.245
	Loss of smell	05 (10.6)	01 (7.7)	0.098 *	0.613
	Myalgia	26 (55.3)	9 (69.2)	0.811 *	0.283
	Shortness of breath	30 (63.8)	8 (61.5)	0.023 *	0.562
History	Smoking	05 (10.6)	1 (7.7)	0.098 *	0.613
	Diabetes mellitus	22 (46.8)	11 (84.6)	5.881 *	0.015
	Hypertension	29 (61.7)	8 (61.5)	0.000 *	0.617
	CKD	4 (8.5)	1 (7.7)	0.009 *	0.705
	IHD	18 (38.3)	9 (69.2)	3.937 *	0.047
Investigations	Hb (g%)	13.44 ± 1.50	12.83 ± 1.46	1.29 ^#^	0.199
	WBC (10^3^/mm^3^)	7.44 ± 3.21	8.89 ± 4.34	−1.32 ^#^	0.192
	O_2_ saturation (%)	89.23 ± 4.87	87.3 ± 7.84	0.994 ^#^	0.325
	Platelet (10^3^/mm^3^)	229.7 ± 87.9	206.5 ± 63.4	0.890 ^#^	0.377
	D-Dimer (ng/mL)	74 (45–374)	822 (498–885)	83.0 ^$^	<0.0001
	LDH (IU/L)	383.5 (247–477)	375 (293–375)	296.0 ^$^	0.865
	AST (IU/L)	35.5 (22.72–57.26)	33.49 (19.00–46.95)	262.0 ^$^	0.440
	ALT (IU/L)	26.7 (19.1–54.7)	24.2 (18.25–74.3)	299.0 ^$^	0.907
	ALP (IU/L)	72.14 (29.19	76.14 (23.15)	−0.455 ^$^	0.651
	CRP	24 (51.1)	8 (61.5)	0.449 *	0.363
	CK (U/L)	132 (84–209)	268 (151–768)	170.5 ^$^	0.015
Therapy	Immunomodulators	38 (82.6)	12 (92.3)	0.738 *	0.358
	Oxygen	22 (47.8)	13 (100)	8.312 *	0.003
	Mechanical ventilation	20 (42.6)	11 (84.6)	7.215 *	0.007
	Antiviral medications	41 (89.1)	12 (92.3)	0.112 *	0.603
	Steroids	42 (91.3)	13 (100)	1.213 *	0.359
Duration of hospital stay (days)		11.5 (5.75–17.00)	20.0 (9.0–26.0)	184.5 ^$^	0.03
Duration of O_2_ therapy (days)		11.00 (3.00–16.25)	20.0 (8.0–26.0)	175.5 ^$^	0.019
ICU admission (days)		3.5 (0–9.0)	18.0 (8.5–26.0)	99.5 ^$^	<0.0001

* Chi square value for data expressed as n (%), test of significance Chi square test; ^#^
*t* value for data expressed as means ± SD, test of significance Student’s unpaired t-test; ^$^ Mann–Whitney U for data expressed as medians (interquartile range), test of significance Mann–Whitney U test.

**Table 2 jpm-12-01729-t002:** Comparison of cytokines and chemokines in the subgroups of the COVID-19 patients.

Special Parameters *	Survivor (*n* = 47)	Non-Survivor (*n* = 13)	Mann–Whitney-U	*p* Value
IL-1 β (pg/mL)	148.85 (94.01–433.40)	50.71 (26.32–106.80)	121.0	0.001
IL-2 (pg/mL)	188.5 (42.01–362.88)	44.10 (22.71–67.25)	164.0	0.011
LTB-4 (pg/mL)	44.95 (12.70–249.79)	11.12 (8.58–113.12)	199.0	0.056
CCL-2 (ng/mL)	675.26 (475.05–974.58)	563.26 (448.2–943.95)	278.0	0.622
IL-8 (pg/mL)	282.49 (467.44–322.17)	168.48 (614.06–304.56)	259.0	0.404
IL-10 (pg/mL)	4.54 (2.12–5.83)	2.3 (1.89–6.03)	253.5	0.351

* Data expressed as medians (interquartile ranges), test of significance applied Mann–Whitney U test. IL, interleukin; LTB, leukotriene B.

**Table 3 jpm-12-01729-t003:** Comparison of cytokine and chemokines levels between subjects with and without COVID.

Parameter	Control (*n* = 60)	COVID-19+ (*n* = 60)	Survivors (*n* = 47)	Non-Survivors (*n* = 13)	*X* ^2^	*p* Value
IL-1 β (pg/mL)	3.5 (0.4–5.1)	121.35 (53.43–288.05) ^a^	148.85 (94.01–433.40) ^a^	50.71 (26.32–106.80) ^a,b,c^	11.916	<0.0001
IL-2(pg/mL)	1.7 (1.3–2.1)	144.12 (40.01–338.60) ^a^	188.5 (42.01–362.88) ^a^	44.10 (22.71–67.25) ^a,b,c^	6.907	<0.0001
LTB-4 (ng/mL)	1.36 (1.18–1.83)	37.70 (11.13–229.56) ^a^	44.95 (12.70–249.79) ^a^	11.12 (8.58–113.12) ^a^	3.213	<0.0001
CCL-2 (pg/mL)	9.35 (7.33–15.19)	673.57 (457.72–965.61) ^a^	675.26 (475.05–974.58) ^a^	563.26 (448.2–943.95) ^a^	0.215	<0.0001
IL-8 (pg/mL)	14.1 (0.4–20.00)	2381 (467–3177) ^a^	2824 (467–3221) ^a^	1689 (614–3045) ^a,b,c^	2.152	<0.0001
IL-10 (pg/mL)	3.6 (3.1–4.00)	3.88 (1.98–5.80)	4.54 (2.12–5.83)	2.3 (1.89–6.03)	0.985	0.805

All values expressed as Median (Range) IL, interleukin; LT, leukotriene. a: *p*-value < 0.05 versus Controls, b: *p*-value < 0.05 versus COVID-19+, c: *p*-value < 0.05 versus Survivors.

**Table 4 jpm-12-01729-t004:** ROC curve analysis of possible predictors of mortality.

Variables	Area	Std. Error	*p* Value	95% CI		Cut-Off	Sensitivity (%)	Specificity (%)
				Low	High			
Age (years) *	0.787	0.084	0.002	0.623	0.952	60	84.6	66.0
WBC (10^3^/mm^3^) *	0.597	0.085	0.286	0.431	0.764	6.15	84.6	40.4
ALP (IU/L) *	0.565	0.087	0.473	0.396	0.735	66	76.9	42.6
D-dimer (ng/mL) *	0.864	0.057	0.000	0.752	0.976	151	92.3	68.1
CK (U/L) *	0.721	0.089	0.015	0.546	0.896	142.5	84.6	61.7
O_2_ saturation on room air (%) ^#^	0.608	0.115	0.298	0.383	0.833	87.5	50	73.7
Platelet (10^3^/mm^3^) ^#^	0.483	0.104	0.869	0.279	0.686	270	90	76.3
LDH (IU/L) ^#^	0.589	0.103	0.388	0.388	0.791	0.7	42.1	57.9
AST (IU/L) ^#^	0.605	0.103	0.310	0.403	0.808	48.6	90	34.2
ALT (IU/L) ^#^	0.545	0.111	0.666	0.327	0.762	33.55	80	42.1
IL-1 β (pg/mL) ^#^	0.802	0.069	0.001	0.667	0.937	109.46	84.6	68.1
IL-2 (pg/mL) ^#^	0.732	0.070	0.011	0.594	0.870	85.54	84.6	70.2
LTB-4 (ng/mL) ^#^	0.674	0.084	0.056	0.510	0.838	11.13	53.8	83
CCL-2 (pg/mL) ^#^	0.545	0.094	0.622	0.361	0.729	467.21	46.2	76.6
IL-8 (pg/mL) ^#^	0.576	0.075	0.404	0.429	0.723	3081	92.3	38.3
IL-10 (pg/mL) ^#^	0.585	0.093	0.351	0.402	0.768	3.31	69.2	63.8

* Values greater than cut-off show higher risk for mortality; ^#^ Values less than cut-off show higher risk for mortality.

## Data Availability

The datasets used and/or analyzed in this study are available from the corresponding author on reasonable request.

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
