# Peer review of "IL-2 and IL-1β Patient Immune Responses Are Critical Factors in SARS-CoV-2 Infection Outcomes"

_jpm, 2022, doi:10.3390/jpm12101729_

Round 1

Reviewer 1 Report

The authors proposed and demonstrated the immune disorders encountered in patients with COVID 19 and their significance in terms of survival.

The methods are adequate and the results are explicit.

My recommendations are

- compliance with the instructions for authors, especially regarding abbreviations, tables, bibliography

The study has statistically significant results, but I am not sure that 60 patients are enough to generalize the conclusions or the therapeutic intervention.

Author Response

- We did our level best to follow authors' instructions (MDPI Style Guide). If there is specific table, abbreviation, or bibliography, we are ready to fix.

- In our prospective cohort study, observational, we dust described what we have found in our population of hospitalized COVID-19 patients. Some variables under the study were significant other were not. Of course, with this sample size and single center study, we could not claim generalizability of our findings. In addition, we did not endorse any therapeutic intervention.  

Reviewer 2 Report

Minor comment:

1) Typo: 10 kD interferon-gamma inducible protein (IP-10).  (line 45)

2) Overall, you suggested that changes in the immune system, including changes in cytokine levels, particularly IL-1β and IL-2, are important factor for determining survival of COVID-19 patient. You also suggested T cells are the main causative immune cell type. A comparison of peripheral immune cell type including T cells by simple staining between survivors and non-survivors among elderly COVID-19 patients, would better support your hypothesis.

Author Response

Comments reply:

1) The mentioned typo (line 45) was corrected

2) Thanks for summarizing our study findings clearly. Of course, your valuable suggestion will be taken in consideration in our future similar project to support the "immunological collapse" hypothesis.